# More emergency patients presenting with chest pain

**Tim Alex Lindskou**[1]*, **Patricia Jessen Andersen**[2], **Erika Frischknecht Christensen**[1,3], **Morten Breinholt Søvsø**[1]

**1** Centre for Prehospital and Emergency Research, Aalborg University and Aalborg University Hospital, Aalborg, Denmark, **2** Department of Cardiology, Aalborg University Hospital, Aalborg, Denmark, **3** Clinic of Internal and Emergency Medicine, Department of Emergency and Trauma Care, Aalborg, Denmark

* tim.l@rn.dk

## Abstract

### Introduction

Throughout recent years the demand for prehospital emergency care has increased significantly. Non-traumatic chest pain is one of the most frequent complaints. Our aim was to investigate the trend in frequency of the most urgent ambulance patients with chest pain, subsequent acute myocardial infarction (AMI) diagnoses, and 48-hour and 30-day mortality of both groups.

### Methods

Population-based historic cohort study in the North Denmark Region during 2012–2018 including chest pain patients transported to hospital by highest urgency level ambulance following a 1-1-2 emergency call. Primary diagnoses (ICD-10) were retrieved from the regional Patient Administrative System, and descriptive statistics (distribution, frequency) performed. We evaluated time trends using linear regression, and mortality (48 hours and 30 days) was assessed by the Kaplan Meier estimator.

### Results

We included 18,971 chest pain patients, 33.9% (n = 6,430) were diagnosed with"Diseases of the circulatory system" followed by the non-specific R- (n = 5,288, 27.8%) and Z-diagnoses (n = 3,634; 19.2%). AMI was diagnosed in 1,967 patients (10.4%), most were non-ST-elevation AMI (39.7%). Frequency of chest pain patients and AMI increased 255 and 22 patients per year respectively, whereas the AMI proportion remained statistically stable, with a tendency towards a decrease in the last years. Mortality at 48 hours and day 30 in chest pain patients was 0.7% (95% CI 0.5% to 0.8%) and 2.4% (95% CI 2.1% to 2.6%).

### Conclusions

The frequency of chest pain patients brought to hospital during 2012–2018 increased. One-tenth were diagnosed with AMI, and the proportion of AMI patients was stable. Almost 1 in of 4 high urgency level ambulances was sent to chest pain patients. Only 1 of 10 patients

**Data Availability Statement:** Data cannot be shared publicly as it contains sensitive patient information. For researchers who have obtained The North Denmark Region approved for the handover of prehospital medical records (https://

rn.dk/sundhed/til-sundhedsfaglige-og-samarbejdspartnere/forskning/videregivelse-af-patientjournaloplysninger only available in Danish), prehospital data can be requested from https://dpv.rn.dk/forskning-og-uddannelse/forskning-og-projekter/ppj (only available in Danish). In-hospital data can be requested from The Danish health data Authority at https://sundhedsdatastyrelsen.dk/da/english/health_data_and_registers/research_services/before_applying.

**Funding:** The authors received no specific funding for this work.

**Competing interests:** The authors have declared that no competing interests exist.

with chest pain had AMI, and overall mortality was low. Thus, monitoring the number of chest pain patients and AMI diagnoses should be considered to evaluate ambulance utilisation and triage.

## Introduction

Emergency care plays a critical role in general healthcare, both in the emergency departments (ED) and the prehospital emergency medical services. Throughout recent years there has been a significant increase in emergency ambulance dispatches and in the demand for prehospital emergency care. The increasing demand challenges several countries in the Western world with Denmark being no exception [1, 2]. Increased life expectancy, increased comorbidity, and also easier access to healthcare, around the clock, has been suggested as possible causes [2–4].

Non-traumatic chest pain persists as one of the most frequent chief complaints in emergency departments in various countries [5–7]. In the North Denmark Region, recent studies showed that new onset chest pain appeared as the sixth and seventh most common complaint among 1-1-2 emergency calls and emergency dispatches [8, 9]. In 2010, the National Board of Health released a consolidation act with the intent to increase the area of focus on early detection of acute coronary syndrome (ACS), as patients with ACS in 20% of the cases present differently, e.g., with dyspnoea, diffuse pain in upper limb areas, or nausea [10]. One could hypothesize that the effect of this consolidation act was that patients calling the emergency number due to chest pain, will have an even higher likelihood of having an ambulance dispatched. Moreover, the symptom chest pain may encompass several severe underlying conditions, and studies have showed that less than one in five patients presenting with chest pain, have ACS [11, 12]. Yet, most patients calling the emergency number with chest pain are triaged with the highest level of urgency [13].

Although previous population-based studies have examined the overall diagnostic pattern and outcome for emergency ambulance patients [14, 15], most research on prehospital care has been focusing on life-threatening emergencies requiring immediate help, including out-of-hospital cardiac arrest [15–18] or "acute myocardial infarction" (AMI) [19]. Furthermore, studies of emergency medical services have more frequently assessed patient characteristics and symptomatology, and have less frequently focused on hospital diagnoses according to the International Statistical Classification of Diseases and Related Health Problems 10th Revision (ICD-10) [20–22].

With an increasing number of ambulance dispatches, and the high frequency of patients with chest pain, our aim was to investigate the trend in frequency of the highest urgency level ambulance patients with the symptom chest pain over time and whether there was a corresponding trend in patients diagnosed with AMI. Moreover, we aimed to describe 48-hours and 30-day mortality in patients presenting with chest pain as chief complaint.

## Methods

### Study design and data sources

We performed a registry-based cohort study of ambulance patients with the symptom chest pain, who was brought to a hospital.

## Study setting

In Denmark, healthcare, including prehospital emergency medical services, is tax-funded, with free-of-charge access to ambulance services. All ambulance services operate according to obligations contracted with one the five individual Danish healthcare regions. The police receive all emergency calls. If it is of medical nature, the call is forwarded to regional Emergency Medical Coordination Centres (EMCC), where healthcare professionals receive the call. The healthcare professionals assess the severity and/or need for ambulance or first lay responders. When doing so, they use a criteria-based dispatch decision support tool, the Danish Index for Emergency care (Danish Index) guiding them in terms of which level of urgency and which response to send [23]. This system encompasses 37 criteria: each representing symptoms of or a potential life-threatening condition with varying level of urgencies. Urgencies are ranked from level A to E, with urgency level A being potentially life-threatening; ambulance run with lights and sirens. Level B is urgent, but not life-threatening; level C is non-acute, but requires observation in ambulance or at hospital; level D is ambulance dispatch with patient in bed; level E is transport to hospital by other means, i.e., taxi, referral to primary healthcare provider or self-care after advice [24]. As an example, chest pain, the 10th criteria in Danish Index, has several conditions prompting an A level response, e.g., "newly arisen strong pains in the centre of the chest for more than five minutes" and "chest pain or discomfort and breathing difficulties" [23].

The North Denmark Region currently has approximately 592,000 inhabitants, (580,000 in 2012 and 590,000 in 2018) [25] and is comprised of both urban and rural areas. One organization is responsible for all emergency medical services in the region. Since April 2006, the region has been using an electronic prehospital medical record in all prehospital units. In 2015, the system was updated and implemented nationwide.

## Study population

We included all patients to whom an ambulance was sent as urgency level A due to chest pain according to the Danish Index criteria, as assessed at the emergency call, and who were subsequently brought to a hospital.

Patients were included in the periods 2012 to 2014 as well as 2016 to 2018 (2015 excluded due to transition to new medical record systems). Patients with more than one ambulance run in the study periods were included.

We obtained the diagnosis related to the primary reason for hospital contact according to ICD-10, from the regional Patient Administrative System. Data on Danish Index criteria for ambulance dispatch were retrieved from the logistic system, Logis CAD (Logis Solutions A/S, Nærum, Denmark). Finally, The Danish Civil Registration System provided information regarding age, sex, and the patients vital status (i.e., deceased).

## Endpoints

Primary endpoint for this study was the frequency of patients with the specific complaint "chest pain", the diagnoses received in hospital (at ICD-10 chapter and subcategory level), especially the frequency of AMI and related diagnoses. Secondary endpoint was outcome in terms of 48-hours and 30-day mortality rates.

## Ethics

The Danish Patient Safety Authority approved the study for the handover of medical records (ID 3-3013-1675/3). Furthermore, the study was registered in the North Denmark Region's list

of ongoing projects (ID 2020–026). According to Danish legislation, no patient consent or further approval (e.g., by an ethics committee), is required when approval for the handover of patient medical records has been given.

### Statistical analysis

Data was anonymized for analysis.

Descriptive statistics was used to assess the distribution of diagnoses according to ICD-10 as frequencies and numbers. Mean and standard deviation was used for normally distributed data both for primary and secondary endpoints. Additionally, the distribution of most frequent primary diagnoses in patients presenting with chest pain and subcategory diagnoses of acute myocardial infarction, respectively, were reported as absolute numbers.

We used linear regression to assess the trend in frequency of both ambulance dispatches due to chest pain and patients diagnosed with acute myocardial infarction.

We used the Kaplan-Meier estimator to report 48 hour- and 30-day mortality rate estimates with 95% confidence intervals. In cases where the patient had more than one ambulance run in the study period, we included a sensitivity analysis, one using only the patient first contact and one using their last contact.

Statistical analyses were performed with Stata/MP 17.0, revision 19 Jul 2022 (StataCorp LCC, 4905 Lakeway Drive, College Station, Texas 77845 USA).

## Results

In the study period there were 77,604 patients, to whom an ambulance was dispatched with urgency level A and brought the patient to a hospital. Of these 18,971 (24.4%) were due to chest pain at the emergency call (Fig 1), and 1,969 (10.4%) were subsequently diagnosed with AMI in hospital.

### Patient characteristics and diagnostic pattern

The most frequent used ICD-10 main chapter was "Diseases of the circulatory system", accounting for a third of patients (n = 6,430, 33.9%). This was followed by the non-specific "Symptoms and signs" which accounted for a fourth (5,288, 27.8%), and "Other factors" with a fifth (3,634, 19.2%) of all patients.

The single most used specific diagnosis for patients presenting with chest pain at emergency calls was R07.4, "chest pain, unspecified" (3,262), and was followed by Z03.9 "Observation for suspected disease or condition, unspecified" (2,014). Patient age and distribution of sex varied according to diagnoses (Table 1).

Most patients with an AMI diagnosis were specifically diagnosed with non-ST-elevation myocardial infarction (I21.4) comprising 39.7% and followed by ST-elevation acute myocardial infarction (I21.3) accounting for 25.2%. See S1 Table for distribution of remaining specific diagnoses among AMI patients.

Patients subsequently diagnosed with AMI had a median age of 68 years (range: 31 to 102) and most were men (73.1%) (Fig 2).

### Time trend

During the study period there was a substantial increase in the absolute number of patients with chest pain, corresponding to 255.41 more patients per year (linear regression: p<0.001, 95% CI: 209.50 to 300.93). Likewise, patients with a subsequent AMI diagnosis also increased substantially with 22.07 patients per year (linear regression: p = 0.004, 95% CI: 12.04 to 32.11).

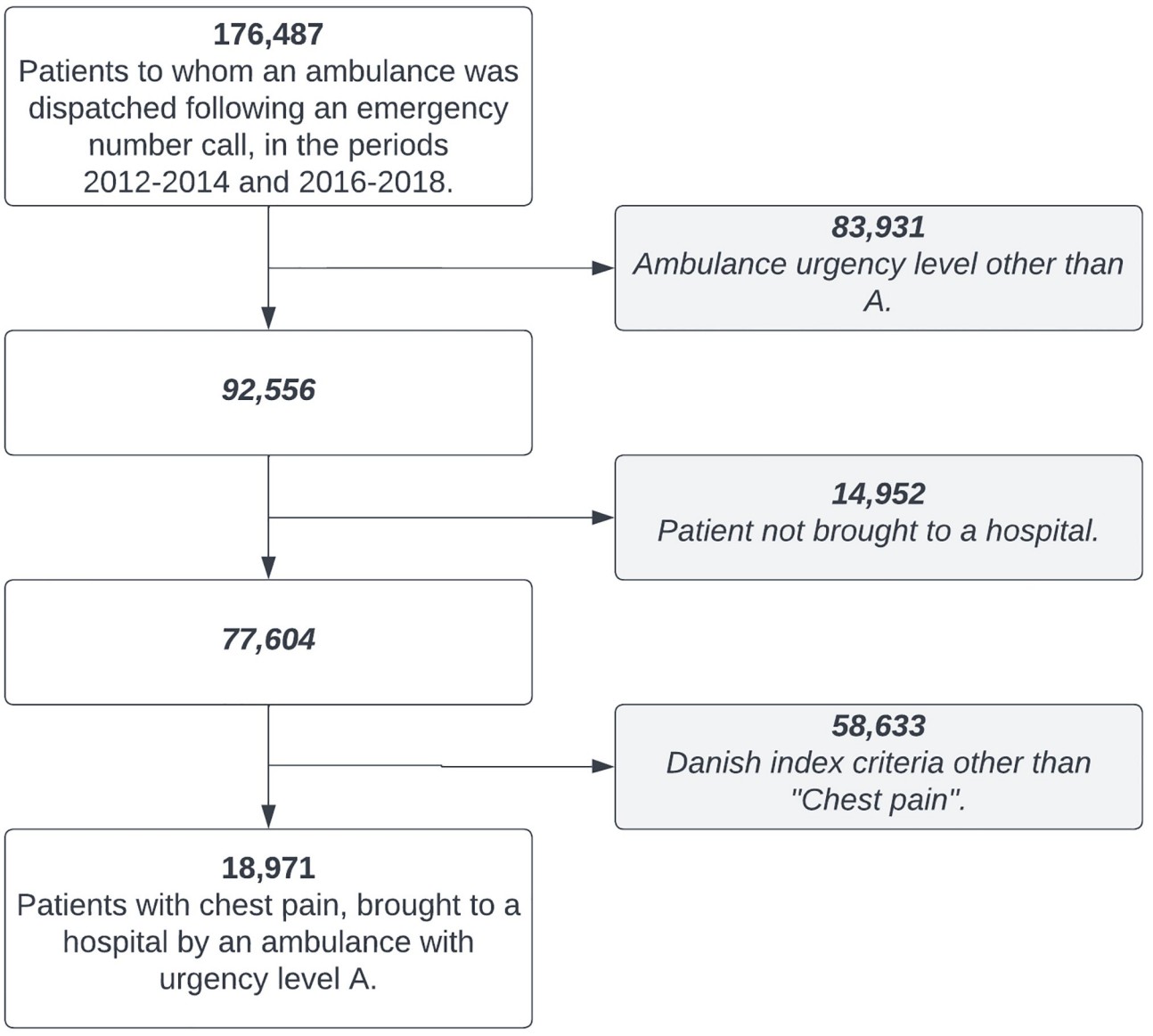

**Fig 1. Flowchart of the included study population.**

However, among the patients with chest pain, the proportion subsequently diagnosed with AMI remained stable, with a tendency towards 0.12 fewer patients per year, although not statistically significant (Table 2) (linear regression: p = 0.15, 95% CI: -0.34 to 0.07).

## Mortality

For all included patients, the overall 48-hour mortality was 0.6% (95% CI 0.5% to 0.7%) and 30-day mortality was 2.4% (95% CI 2.2% to 2.6%). The 48-hour mortality ranged from 0.5 to 0.8%; and 30-day mortality from 2.1 to 2.9%.

For patients subsequently diagnosed with AMI, the overall 48-hour mortality was 1.1% (95%CI 0.8% to 1.8%), and 30-day mortality was 4.4% (95% CI 3.6% to 5.4%). The 48-hour mortality ranged from 0.8% to 1.4%, whereas the 30-day mortality ranged from 4.2% to 5.0%.

**Table 1. Patient characteristics within the ten most frequent specific hospital diagnoses and corresponding main diagnostic chapters (ICD-10) following 1-1-2 emergency calls with chief symptom of chest pain.**

| ICD-10 diagnoses | Frequency, n (%) | Age (range) | Female (%) |
|---|---|---|---|
| Total | 18,971 (100) | 65 (0–104) | 42.8 |
| **ICD-10 main chapter: Diseases of the circulatory system** | **6,430 (33.9)** | **70 (0–104)** | **36.6** |
| I20.9: Angina pectoris, unspecified | 982 (5.2) | 70 (23–101) | 39.8 |
| I21.4: Non-ST-elevation acute myocardial infarction | 807 (4.5) | 71 (31–102) | 30.2 |
| I48.9: Atrial fibrillation or flutter, unspecified | 588 (3.1) | 73 (20–97) | 53.2 |
| I21.3: ST-elevation acute myocardial infarction | 553 (2.9) | 64 (35–98) | 24.1 |
| I25.9: Chronic ischaemic heart disease, unspecified | 402 (2.2) | 70 (37–95) | 28.7 |
| **ICD-10 main chapter: Symptoms and signs** | **5,288 (27.9)** | **59 (1–102)** | **46.6** |
| R07.4: Chest pain, unspecified | 3,262 (17.2) | 58 (8–102) | 44.9 |
| R07.3: Other chest pain | 363 (1.2) | 56 (15–99) | 51.8 |
| **ICD-10 main chapter: Other factors** | **3,634 (19.1)** | **61 (18–93)** | **46.6** |
| Z03.9: Observation for suspected disease or condition, unspecified | 2,014 (10.6) | 62 (17–99) | 48.0 |
| Z03.5: Observation for other suspected cardiovascular diseases | 624 (3.3) | 60 (15–99) | 44.7 |
| Z03.4: Observation for suspected myocardial infarction | 430 (2.3) | 60 (16–101) | 43.0 |
| **ICD-10 main chapter: Respiratory diseases** | **1,072 (5.7)** | **74(15–102)** | **42.1** |
| J18.9: Pneumonia, unspecified | 460 (2.4) | 75 (17–102) | 41.1 |
| **Remaining ICD-10 main chapters** | **2,547 (13.4)** | **74 (15–102)** | **42.1** |

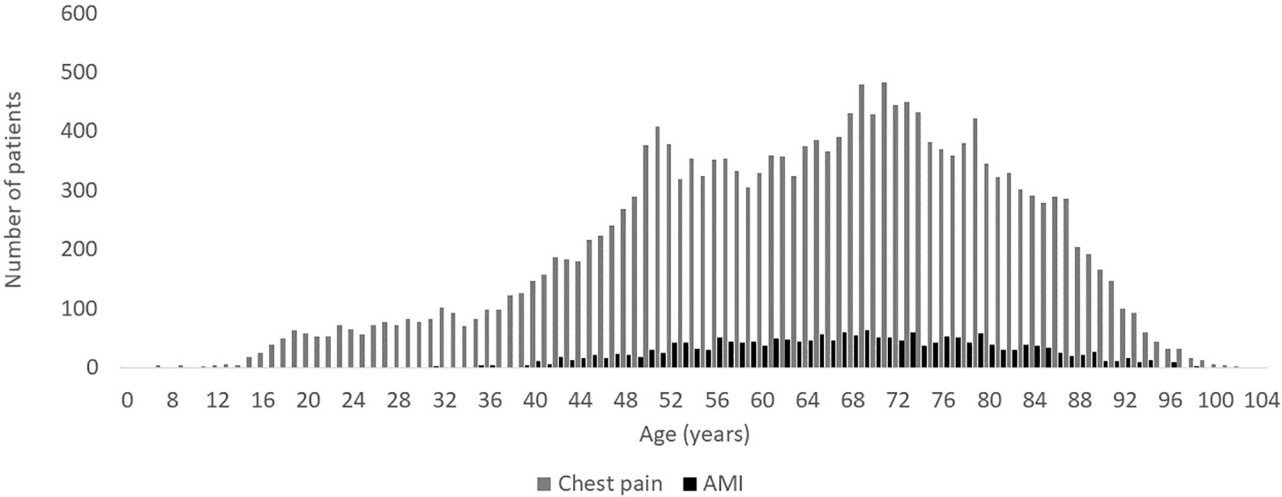

**Fig 2. Age distribution among all patients with chest pain according to Danish Index (grey) and patients subsequently diagnosed with AMI (black), in the years 2012–2018 (excl. 2015).**

**Table 2. Trends in number of patients with chest pain as main complaint and subsequently acute myocardial infarction diagnosis.** AMI, acute myocardial infarction.

| Year | 2012 | 2013 | 2014 | 2016 | 2017 | 2018 |
|---|---|---|---|---|---|---|
| **Patients with chest pain, frequency n (proportion of all urgency level A patients, %)** | 2,430 (22.60) | 2,564 (23.26) | 2,968 (24.66) | 3,465 (25.53) | 3,559 (23.94) | 3,983 (25.96) |
| **Patients with subsequent AMI, frequency n (proportion of level A patients with chest pain, %)** | 264 (10.86) | 261 (10.18) | 327 11.02) | 367 (10.60) | 358 (10.06) | 392 (9.84) |

**Table 3. Cumulative deaths and mortality estimates for patients with chest pain, and those with a subsequent AMI diagnosis.** Cumulative number of deaths within 48-hours omitted due to microdata. AMI, acute myocardial infarction.

| Year | 2012 | 2013 | 2014 | 2016 | 2017 | 2018 | Overall |
|---|---|---|---|---|---|---|---|
| Number of patients with chest pain, n | 2,430 | 2,565 | 2,968 | 3,465 | 3,560 | 3,983 | 18,971 |
| 48-hour mortality, % (95% CI) | 0.6 (0.4–1.0) | 0.7 (0.4–1.1) | 0.8 (0.5–1.2) | 0.6 (0.4–0.9) | 0.5 (0.3–0.8) | 0.6 (0.4–0.9) | 0.6 (0.5–0.7) |
| Cumulative number of deaths day 30, n | 63 | 74 | 76 | 72 | 82 | 85 | 452 |
| 30-day mortality, % (95% CI) | 2.6 (2.0–3.3) | 2.9 (2.3–3.6) | 2.6 (2.1–3.2) | 2.1 (1.7–2.6) | 2.3 (1.9–2.9) | 2.1 (1.7–2.6) | 2.4 (2.2–2.6) |
| Year | 2012 | 2013 | 2014 | 2016 | 2017 | 2018 | Overall |
| Number of patients with subsequent AMI, n | 264 | 261 | 327 | 367 | 358 | 392 | 1,969 |
| 48-hour mortality, % (95% CI) | 0.76 (0.2–3.0) | 1.2 (0.4–3.5) | 1.5 (0.6–3.6) | 1.4 (0.6–3.20) | 1.1 (0.4–3.0) | 1.0 (0.4–2.7) | 1.1 (0.8–1.8) |
| Cumulative number of deaths day 30, n | 11 | 13 | 13 | 18 | 15 | 17 | 87 |
| 30-day mortality, % (95% CI) | 4.2 (2.3–7.4) | 5.0 (2.9–8.4) | 4.0 (2.3–6.8) | 4.9 (3.1–7.7) | 4.2 (2.6–6.9) | 4.3 (2.7–6.9) | 4.4 (3.6–5.4) |

Overall, there was little variation in mortality during the study period years (Table 3).

For all included patients, those diagnosed within the respiratory diseases chapter had the highest mortality: This mortality was significantly higher than for those diagnosed within the circulatory diseases main chapter (S2 Table).

For the specific diagnoses with the highest number of deaths, cardiac arrest had the highest mortality, followed by acute respiratory failure, and aortic aneurysm and dissection (S3 Table).

### Sensitivity analysis

Including only the patients' first contact, in cases where they had more than one ambulance run in the study period, did not change the trend in mortality during the study period (Table 4).

The same tendency was present when only including the patients' last contact, albeit the mortality was overall higher (Table 5).

## Discussion

### Key results

Over a six-year period, we found an increase in the number of patients calling the emergency number due to chest pain and who were subsequently brought to a hospital with an urgency

**Table 4. Cumulative deaths and mortality estimates for patients with chest pain, and those with a subsequent AMI diagnosis.** Note for patients with more than one ambulance contact in the study periods, only the first contact was used. Cumulative number of deaths within 48-hours omitted due to microdata. AMI, acute myocardial infarction.

| Only first contact included | | | | | | | |
|---|---|---|---|---|---|---|---|
| Year | 2012 | 2013 | 2014 | 2016 | 2017 | 2018 | Overall |
| Number of patients with chest pain | 2,141 | 2,105 | 2,279 | 2,708 | 2,596 | 2,772 | 14,601 |
| 48-hour mortality, % (95% CI) | 0.7 (0.4–1.1) | 0.8 (0.5–1.2) | 0.9 (0.6–1.4) | 0.6 (0.4–1.0) | 0.5 (0-3-0.9) | 0.6 (0.4–0.9) | 0.7 (0.5–0.8) |
| Cumulative number of deaths day 30, n | 55 | 58 | 61 | 51 | 62 | 57 | 344 |
| 30-day mortality, % (95% CI) | 2.6 (2.0–3.3) | 2.8 (2.1–3.6) | 2.7 (2.1–3.4) | 1.9 (1.4–2.5) | 2.4 (1.9–3.1) | 2.1 (1.6–2.7) | 2.4 (2.1–2.6) |
| Year | 2012 | 2013 | 2014 | 2016 | 2017 | 2018 | Overall |
| Number of patients with subsequent AMI | 255 | 243 | 290 | 328 | 317 | 348 | 1781 |
| 48-hour mortality, % (95% CI) | 0.8 (0.2–3.1) | 0.8 (0.2–3.3) | 1.4 (0.5–3.6) | 1.2 (0.5–3.2) | 1.3 (0.5–3.3) | 1.2 (0.4–3.0) | 1.1 (0.7–1.7) |
| Cumulative number of deaths day 30, n | 11 | 10 | 11 | 13 | 14 | 14 | 73 |
| 30-day mortality, % (95% CI) | 4.3 (2.4–7.7) | 4.1 (2.2–7.5) | 3.8 (2.1–6.7) | 4 (2.3–6.7) | 4.4 (2.6–7.3) | 4 (2.4–6.7) | 4.1 (3.3–5.1) |

**Table 5. Cumulative deaths and mortality estimates for patients with chest pain, and those with a subsequent AMI diagnosis.** Note for patients with more than one ambulance contact in the study periods, only the last contact was used. Cumulative number of deaths within 48-hours omitted due to microdata. AMI, acute myocardial infarction.

| Only last contact included | | | | | | | |
|---|---|---|---|---|---|---|---|
| **Year** | **2012** | **2013** | **2014** | **2016** | **2017** | **2018** | **Overall** |
| Number of patients with chest pain | 1,659 | 1,837 | 2,214 | 2,647 | 2,817 | 3,427 | 14,601 |
| 48-hour mortality, % (95% CI) | 0.9 (0.6–1.5) | 1.0 (0.6–1.6) | 1.1 (0.7–1.6) | 0.8 (0.5–1.2) | 0.6 (0.4–1.0) | 0.7 (0.5–1.0) | 0.8 (0.7–1.0) |
| Cumulative number of deaths day 30, n | 58 | 71 | 74 | 68 | 78 | 82 | 431 |
| 30-day mortality, % (95% CI) | 3.5 (2.7–4.5) | 3.9 (3.1–4.9) | 3.3 (2.7–4.2) | 2.6 (2.0–3.3) | 2.8 (2.2–3.4) | 2.4 (1.9–3.0) | 3.0 (2.7–3.2) |
| **Year** | **2012** | **2013** | **2014** | **2016** | **2017** | **2018** | **Overall** |
| Number of patients with subsequent AMI | 196 | 196 | 260 | 305 | 293 | 357 | 1,607 |
| 48-hour mortality, % (95% CI) | 1.0 (0.3–4.0) | 1.5 (0.5–4.67) | 1.9 (0.8–4.6) | 1.6 (0.7–3.9) | 1.37 (0.5–3.6) | 1.1 (0.4–3.0) | 1.4 (1.0–2.2) |
| Cumulative number of deaths day 30, n | 10 | 12 | 12 | 16 | 14 | 16 | 80 |
| 30-day mortality, % (95% CI) | 5.1 (2.8–9.3) | 6.1 (3.5–10.5) | 4.6 (2.7–8.0) | 5.3 (3.3–8.4) | 4.8 (2.9–7.9) | 4.5 (2.8–7.2) | 5.0 (4.0–6.2) |

level A ambulance. Furthermore, the number of patients who received an AMI diagnosis also increased. However, the proportion of AMI among chest pain patients tended towards a decrease at the end of the study period.

Mortality did not increase significantly during the study period.

## Comparison with other studies

Internationally, McDevitt-Petrovic et al. [6] conducted a study in a Northern-Irish setting assessing the frequency of patients presenting with chest pain at the ED and the subsequent non-cardiac chest pain diagnoses during the years 2013 to 2016. They found an increase in chest pain presentations from 2,381 to 3,239, and an increase in the frequency of chest pain presentations compared to non-chest pain (4.1% to 5.1%).

Although the current study only included urgency level A ambulance patients brought to a hospital, we also found an increase in patients, from 2,430 to 3,983 (22.60 to 25.96% of all urgency level A dispatches).

The current study only included level A urgency dispatches to include patients with the highest likelihood of AMI, as patients with acute chest pain are most often assigned level A urgency at time of dispatch.

A Danish study by Pedersen et al. from the Central Denmark Region during 2015–2016 included patients who had an ambulance dispatched both following an emergency number call (urgency levels A, B, and C) and by request from a general practitioner. Despite the difference in inclusion criteria, their prevalence of AMI of 10.5%, aligns with our findings [13].

A recent study investigated all acute somatic hospital contacts in Denmark in the period 2016–2018. Amongst others, the study found 25.4–28.1% of hospitalizations had an non-specific diagnosis, either R "symptoms and signs" or Z "other factors" [26]. In the current study, among patients calling the emergency number with chest pain, the combined non-specific R and Z diagnoses, accounted for nearly half (47%) of the patients. In comparison with all chest pain patients, the subset with AMI were older and much more often male (73.1% vs 57.2%). Our results suggest that prehospital chest pain, may in part contribute to the high number of patients with non-specific diagnoses in-hospital.

## Mortality

In a study from the North Denmark Region during 2016–2018, Ibsen et al. [8] found that the symptom of "chest pain" according to Danish Index, had a relatively low overall 30-day

mortality of 2.1%. This corresponds to the 30-day mortality of 2.4% in the current study, despite the inclusion of level A dispatches only, where the patients are expected to be in a more severe acute medical condition.

Ibsen et al. also found the 30-day mortality for chest pain was in fact was four to six times less deadly than the symptom "breathing difficulty" or "unclear problem". Chest pain was the seventh most common complaint preceding an emergency dispatch in that study. Compared to our study, this suggest the symptom "breathing difficulties" is linked to a greater danger than "chest pain".

As in the study by Pedersen et al, the 30-day mortality among patients subsequently diagnosed with AMI in the current study is relatively low with 4.1% [13]. The low mortality might be explained by the prehospital setup with primary percutaneous coronary intervention ideally bypassing the emergency department and leading to faster treatment [27].

## Strengths and limitations of the study

One major strength of our study is its population-based design; thus, it includes all patients, assessed as having chest pain at the emergency call, who were subsequently brought to an hospital by an ambulance dispatched as level A urgency. The linkage between registries through the unique civil registration number of each patient reduced loss-to-follow-up. As mentioned in the method section, we only included patients with urgency level A, to achieve the highest likelihood of AMI patients. Thus, AMI patients transported with urgency level B have not been included in our study. This may have led to an underestimation of AMI prevalence and could affect our mortality estimates in both directions. Likewise, we only included patients brought to a hospital. This could introduce a bias in the prevalence of the patients, as well as skewed mortality towards either an under- or overestimation since we know the number of non-conveyed patients in the North Denmark region is approximately 10% of all ambulance runs [15].

We have no information on diagnostics such as electrocardiograms, blood samples or coronary angiography in patients receiving a subsequent AMI diagnosis. We also did not consider comorbidity in our mortality analysis. Further, we only assessed patients presenting with chest pain, and therefore, we miss out on patients presenting atypically, such as atypical pain and dyspnoea, like mentioned in consolidation act regarding diagnosis of ACS released by the National Board of Health in 2010 [10]. As discussed previously, the symptom "breathing difficulties" or "dyspnoea" has been associated with a higher mortality [28] than chest pain despite being a frequent cause of 1-1-2 emergency calls [8].

Patients with more than one ambulance run were included, since every time a patient has a new contact at the hospital a new assessment is performed. However, we chose to perform our mortality analyses based on only the first patient contact, which may have introduced a possibility of underestimating the mortality.

## Conclusions

In this population-based study in the North Denmark region we found an increase in patients brought to hospital with chest pain as chief complaint throughout the years 2012–2018. One-tenth were subsequently diagnosed with AMI, and the proportion of patients with AMI was stable with a small tendency towards a decrease at the end of the study period.

Compared to mortality among other symptoms at the emergency call, chest pain does not seemingly appear as severe as might have been expected. Additionally, among patients with chest pain, nearly half subsequently received a non-specific diagnosis, not related to manifest cardiac disease.

Almost 1 in of 4 emergency ambulance urgency level A are sent to patients presenting with chest pain. Only 1 of 10 patients with chest pain in this study had AMI and the overall mortality was low. Thus, a continuous monitoring of the number of chest pain patients and AMI patients should be considered to evaluate ambulance utilisation.

## Perspectives

Our findings underline the fact that patients calling for an ambulance with chest pain is comprised of around 10% acute myocardial infarctions and a remaining diverse group of patients, where nearly half are assigned a non-specific diagnosis at discharge. We found an increase in the number of patients with chest pain and with AMI. However, the proportion of AMI within the chest pain population remained stable, tending to decrease. As 1 in 4 of all urgency level A ambulances are sent to chest pain patients and the overall mortality is low, perhaps a revision of the current triage is needed. It could be argued that highest urgency level is not necessary for all patients with chest pain. However, finding those in need of urgent intervention is difficult, since telephone triage is limited by no physical contact or measurements/diagnostics, making it difficult to make substantial changes at this point in the care pathway.

A recent study of patients over the age of 18, with chest pain as their main symptom, have suggested models using age, sex, medical history, and symptomology can improve EMS prioritisation, when compared to criteria index-based prioritisation [29]. Implementing prehospital diagnostics using point-of-care biomarker tests has been shown to help rule out acute myocardial infarction on scene [30], perhaps allowing for a different and subacute care pathway for the patient. Increased time spent on scene with the patient with presumed heart condition has also been associated with reduced mortality [31].

The results of this study confirm the need for such interventions and could be of interest to healthcare planners within emergency medical services, especially when taking into consideration the resources allocated to one specific symptom. It is important to investigate and monitor how this prioritization affect other patients needing emergency medical services, as the demand for ambulances keeps increasing.

## Supporting information

**S1 Table. Diagnoses.** Specific diagnoses among patients to whom an ambulance was sent as urgency level A due to chest pain, and who subsequently were diagnosed with AMI.
(DOCX)

**S2 Table. Mortality according to most frequent diagnosis.** Mortality according to most frequent specific diagnoses among patients to whom an ambulance was sent as urgency level A due to chest pain. 48-hour mortality omitted due to low number of deaths (microdata).
(DOCX)

**S3 Table. Mortality according to highest numbers of deaths.** Mortality according to the specific diagnoses with highest numbers of death among patients to whom an ambulance was sent as urgency level A due to chest pain. 48-hour mortality omitted due to low number of deaths (microdata).
(DOCX)

## Author Contributions

**Conceptualization:** Tim Alex Lindskou, Patricia Jessen Andersen, Erika Frischknecht Christensen, Morten Breinholt Søvsø.

**Data curation:** Tim Alex Lindskou, Patricia Jessen Andersen, Morten Breinholt Søvsø.

**Formal analysis:** Tim Alex Lindskou, Patricia Jessen Andersen, Morten Breinholt Søvsø.

**Investigation:** Patricia Jessen Andersen, Morten Breinholt Søvsø.

**Methodology:** Tim Alex Lindskou, Patricia Jessen Andersen, Erika Frischknecht Christensen, Morten Breinholt Søvsø.

**Project administration:** Tim Alex Lindskou, Morten Breinholt Søvsø.

**Supervision:** Tim Alex Lindskou, Erika Frischknecht Christensen, Morten Breinholt Søvsø.

**Validation:** Erika Frischknecht Christensen.

**Visualization:** Erika Frischknecht Christensen.

**Writing – original draft:** Tim Alex Lindskou, Patricia Jessen Andersen, Morten Breinholt Søvsø.

**Writing – review & editing:** Tim Alex Lindskou, Patricia Jessen Andersen, Erika Frischknecht Christensen, Morten Breinholt Søvsø.

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
