## [Decision Letter · Decision Letter 0]

10 Jan 2023

PONE-D-22-33621More emergency patients presenting with chest painPLOS ONE

Dear Dr. Lindskou,

Thank you for submitting your manuscript to PLOS ONE. After careful consideration, we feel that it has merit but does not fully meet PLOS ONE’s publication criteria as it currently stands. Therefore, we invite you to submit a revised version of the manuscript that addresses the points raised during the review process.

We look forward to receiving your revised manuscript.

Kind regards,

Saraschandra Vallabhajosyula, MD MSc

Academic Editor

PLOS ONE

3. Please upload a copy of Supplemental Table 1 which you refer to in your text on page 9.

Additional Editor Comments:

I agree with the comments from the reviewers. Please revise accordingly.

Reviewers' comments:

Reviewer's Responses to Questions

**Comments to the Author**

1. Is the manuscript technically sound, and do the data support the conclusions?

Reviewer #1: Yes

Reviewer #2: Partly

2. Has the statistical analysis been performed appropriately and rigorously? 

Reviewer #1: Yes

Reviewer #2: Yes

3. Have the authors made all data underlying the findings in their manuscript fully available?

Reviewer #1: Yes

Reviewer #2: Yes

4. Is the manuscript presented in an intelligible fashion and written in standard English?

Reviewer #1: Yes

Reviewer #2: Yes

5. Review Comments to the Author

Reviewer #1: This original manuscript entitled, “More emergency patients presenting with chest pain”, discusses an essential and timely topic. I commend Lindskou et. al. for their choice of topic and study. I have a few comments for the authors to take into account. I have included the detailed comments as an attachment.

Reviewer #2: In this retrospective study, ‘More emergency patients presenting with chest pain’, the authors discuss an important topic. I commend them for their choice of topic, especially its relevance to how we can use these findings as a scaffold to optimize healthcare services and make it more efficient. I do have a few comments for their consideration that are attached.

6. PLOS authors have the option to publish the peer review history of their article (what does this mean?). If published, this will include your full peer review and any attached files.

Reviewer #1: **Yes: **Aryan Mehta

Reviewer #2: No

---

## [Author Response · Author response to Decision Letter 0]

6 Feb 2023

Response to reviewers 

We thank the reviewers for their constructive comments on our paper and have edited the manuscript to address their concerns. Below, we have answered each query, and all changes to the manuscripts are returned as track changes and included in the rebuttal letter with reference to line numbers in the manuscript.

We are grateful for their contribution to what we see as an improved manuscript.

Best Regards

Tim Alex Lindskou

Reviewer #1:

In this retrospective population-based study, ‘More emergency patients presenting with chest pain’, the authors discuss an important topic. I commend them for their choice of topic especially its relevance to how we can use these findings as a scaffold to optimize healthcare services and make it more efficient. I do have a few comments for their consideration:

1. I urge the authors to proofread the manuscript again and eliminate the referencing errors in the text. There are some places where references are missing and need to be updated accordingly.

 - We have proofed the paper, the references have been checked, and any inconsistencies have been corrected. In this process, we have added the missing supplemental table S1 referenced on page 9.

2. The relevance of the study is lacking in the text. Given that only a small proportion of patients with chest pain had an acute myocardial infarction and only a minority experienced fatal outcomes, it would be prudent to shed light on the fact of how the authors plan or at least suggest the application of these findings to optimize the healthcare delivery process.

 - Thank you for addressing this. We have added a perspectives section and addressed your points within that (please see page 16).

“Perspectives

Our findings underline the fact that patients calling for an ambulance with chest pain is comprised of around 10% acute myocardial infarctions and a remaining diverse group of patients, where nearly half are assigned a non-specific diagnosis at discharge. We found an increase in the number of patients with chest pain and with AMI. However, the proportion of AMI within the chest pain population remained stable, tending to decrease. As 1 in 4 of all urgency level A ambulances are sent to chest pain patients and the overall mortality is low, perhaps a revision of the current triage is needed. It could be argued that highest urgency level is not neces-sary for all patients with chest pain. However, finding those in need of urgent intervention is difficult, since telephone triage is limited by no physical contact or measurements/diagnostics, making it difficult to make substantial changes at this point in the care pathway. 

A recent study of patients over the age of 18, with chest pain as their main symptom, have suggested models using age, sex, medical history, and symptomology can improve EMS prioritisation, when compared to criteria index-based prioritisation (29). Implementing prehospital diagnostics using point-of-care biomarker tests has been shown to help rule out acute myocardial infarction on scene (30), perhaps allowing for a different and subacute care pathway for the patient. Increased time spent on scene with the patient with presumed heart condition has also been associated with reduced mortality (31). 

The results of this study confirm the need for such interventions and could be of interest to healthcare plan-ners within emergency medical services, especially when taking into consideration the resources allocated to one specific symptom. It is important to investigate and monitor how this prioritization affect other patients needing emergency medical services, as the demand for ambulances keeps increasing.”

3. The authors have not sub-classified the ICD-10 diagnosis of the patients with 48-hr and 30-day mortality. I would request them to add this to the text as it would lead to a more comprehensive understanding of the issues at hand.

 - We agree that going into detail with sub-classification of ICD-10 diagnoses would contribute to better understanding, yet as you point out, fatalities are very few especially within 48 hours, and reporting such microdata conflicts with patient confidentiality. However, we have added supplemental tables on 30-day mortality for sub-classification of diagnoses.

Please see supplemental table S2 and S3.

4. The authors have done a great job at explaining the study methodology. However the result section at times is not coherent with the table the authors have given. It would be of great help if author made it more streamlined and thus make it tangible with the text.

 - We agree and have revised table 1 to also include the ICD-10 main chapters, we refer to in the main text and provided the missing supplemental table referenced on page 9 showing subclassifications of acute myocardial infarction diagnoses. Moreover, we have added supplemental tables showing sub-classifications of diagnoses and the corresponding mortality. Please see supplemental tables 2 and 3.

Reviewer #2: 

General Comments: 

This original manuscript entitled, “More emergency patients presenting with chest pain”, discusses an essential and timely topic. I commend Lindskou et. al. for their choice of topic and study. I have a few comments for the authors to take into account. 

 Major Comments: 

1. The choice of outcomes is interesting. Even though the outcomes including the number of ambulance dispatches for chest pain, patients diagnosed with AMI, and 48-hour and 30-day mortality in patients with chest pain are quite descriptive, I think the authors can perform a sub-group analysis comparing relevant outcomes like mortality, length of stay, complications, etc between patients presenting with chest pain vs non-chest pain patients. 

 - We agree that it could be interesting to compare the chest pain versus non-chest pain groups. However, the latter group was not included in our study population (please see flow chart (figure 1)), and we would need further approval to include this group. Thus, such further analysis is beyond the scope of this paper.

We have however included supplementary material and analysis of the mortality for specific diagnoses (Supplemental table S2 and S3)

2. Even though the topic of discussion here is well thought out and timely, I urge the authors to elaborate more on the significance and implications of this study. 

 - Thank you for addressing this. We have added a perspectives section and addressed your points within that. 

“Perspectives

Our findings underline the fact that patients calling for an ambulance with chest pain is comprised of around 10% acute myocardial infarctions and a remaining diverse group of patients, where nearly half are assigned a non-specific diagnosis at discharge. We found an increase in the number of patients with chest pain and with AMI. However, the proportion of AMI within the chest pain population remained stable, tending to decrease. As 1 in 4 of all urgency level A ambulances are sent to chest pain patients and the overall mortality is low, perhaps a revision of the current triage is needed. It could be argued that highest urgency level is not neces-sary for all patients with chest pain. However, finding those in need of urgent intervention is difficult, since telephone triage is limited by no physical contact or measurements/diagnostics, making it difficult to make substantial changes at this point in the care pathway. 

A recent study of patients over the age of 18, with chest pain as their main symptom, have suggested models using age, sex, medical history, and symptomology can improve EMS prioritisation, when compared to criteria index-based prioritisation (29). Implementing prehospital diagnostics using point-of-care biomarker tests has been shown to help rule out acute myocardial infarction on scene (30), perhaps allowing for a different and subacute care pathway for the patient. Increased time spent on scene with the patient with presumed heart condition has also been associated with reduced mortality (31). 

The results of this study confirm the need for such interventions and could be of interest to healthcare plan-ners within emergency medical services, especially when taking into consideration the resources allocated to one specific symptom. It is important to investigate and monitor how this prioritization affect other patients needing emergency medical services, as the demand for ambulances keeps increasing.”

3. In a similar study done by Pedersen et. Al. Done in Central Denmark the way of entrance into the EMS system was categorized as “112-requested” or “GP-requested”. (a) Under GP requested category, the prehospital response is based on the GPs’ clinical assessment rather than dispatch criteria. Excluding this patient strata may underestimate the prevalence of AMI in patients presenting with chest pain. Can the authors please elucidate why did they not consider this? 

(a) Pedersen CK, Stengaard C, Friesgaard K, Dodt KK, Søndergaard HM, Terkelsen CJ, et al. Chest pain in the ambulance; prevalence, causes, and outcome - a retrospective cohort study. Scandinavian Journal of Trauma, Resuscitation and Emergency Medicine. 2019;27(1):84. 

 - Thank you for raising this question. Our study was motivated by the observation of more and more ambulance dispatches following calls to the national emergency number 1-1-2 in our region - with chest pain being one of the symptoms with the highest frequency, which is why we chose to include patients with a 1-1-2 call, and only patients with an urgency level A ambulance to increase the likelihood of including patients with AMI.

Denmark has five regions, each responsible for healthcare and EMS within each region. The organization of healthcare differs a bit depending on the region. Some have a direct number for GP’s and within these further variation exists concerning when the number is used. As such there is a data limitation, and it may not be possible for all regions to differentiate correctly which pathway was used when an ambulance was requested.

4. Can the authors please be consistent in citing and referencing the manuscript? Also, on page 9 of the manuscript there is an error where reference sources cannot be found. Kindly address this.

 - Thank you for bringing this to our attention. We have proofed the paper, the references have been checked, and any inconsistencies have been corrected.

In this process, we have added the missing supplemental table referenced on p9.

5. Can the authors please address the grammatical errors and inconsistencies throughout the manuscript? 

 - We have proofed the paper once more, and any inconsistencies have been corrected.

---

## [Decision Letter · Decision Letter 1]

8 Mar 2023

More emergency patients presenting with chest pain

PONE-D-22-33621R1

Dear Dr. Lindskou,

We’re pleased to inform you that your manuscript has been judged scientifically suitable for publication and will be formally accepted for publication once it meets all outstanding technical requirements.

Kind regards,

Saraschandra Vallabhajosyula, MD MSc

Academic Editor

PLOS ONE

Additional Editor Comments (optional):

Thank you for your responses.

Reviewers' comments:

Reviewer's Responses to Questions

**Comments to the Author**

1. If the authors have adequately addressed your comments raised in a previous round of review and you feel that this manuscript is now acceptable for publication, you may indicate that here to bypass the “Comments to the Author” section, enter your conflict of interest statement in the “Confidential to Editor” section, and submit your "Accept" recommendation.

Reviewer #1: All comments have been addressed

Reviewer #2: All comments have been addressed

2. Is the manuscript technically sound, and do the data support the conclusions?

Reviewer #1: Yes

Reviewer #2: Yes

3. Has the statistical analysis been performed appropriately and rigorously? 

Reviewer #1: Yes

Reviewer #2: Yes

4. Have the authors made all data underlying the findings in their manuscript fully available?

Reviewer #1: Yes

Reviewer #2: Yes

5. Is the manuscript presented in an intelligible fashion and written in standard English?

Reviewer #1: Yes

Reviewer #2: Yes

6. Review Comments to the Author

Reviewer #1: This original manuscript entitled, “More emergency patients presenting with chest pain”, discusses an essential and timely topic. I commend Lindskou et. al. for their choice of topic and study. The authors have successfully addressed all the comments.

Reviewer #2: In this retrospective population-based study, ‘More emergency patients presenting with chest pain’, the authors discuss an important topic. I commend them for their choice of topic especially its relevance to how we can use these findings as a scaffold to optimize healthcare services and make it more efficient. They have addressed all the comments.

7. PLOS authors have the option to publish the peer review history of their article (what does this mean?). If published, this will include your full peer review and any attached files.

Reviewer #1: No

Reviewer #2: No

---

## [Editor Report · Acceptance letter]

14 Mar 2023

PONE-D-22-33621R1 

More emergency patients presenting with chest pain 

Dear Dr. Lindskou:

I'm pleased to inform you that your manuscript has been deemed suitable for publication in PLOS ONE. Congratulations! Your manuscript is now with our production department. 

Kind regards, 

on behalf of

Dr. Saraschandra Vallabhajosyula 

Academic Editor

PLOS ONE